# Methyl Aminolaevulinic Acid versus Aminolaevulinic Acid Photodynamic Therapy of Actinic Keratosis with Low Doses of Red-Light LED Illumination: Results of Long-Term Follow-Up

**DOI:** 10.3390/biomedicines10123218

**Published:** 2022-12-12

**Authors:** Montserrat Fernández Guarino, Diego Fernández-Nieto, Laura Vila Montes, Dario de Perosanz Lobo

**Affiliations:** Hospital Universitario Ramón y Cajal, Instituto de Investigación Sanitaria Irycis, 28034 Madrid, Spain

**Keywords:** photodynamic therapy, red light, short illumination

## Abstract

Photodynamic therapy (PDT) treatment for multiple actinic keratosis (AK) has been found effective when lower doses of red light were used with methyl aminolaevulinic acid (MAL). The aim of this study was to compare the results of lower doses of red light conventional PDT (h-PDT, 16 J/cm^2^) with MAL and aminolaevulinic acid (ALA) in a long-term follow-up. Patients with more than five symmetrical AK on the scalp who were candidates for PDT were selected and divided randomly between MAL and ALA treatment and patients were followed at 3 and 12 months. The responses were assessed by counting the total AK and the AK per patient. Pain and adverse events were also compiled. A total of 46 patients were treated, 24 with MAL, and 22 with ALA. The two groups were comparable at baseline (*p* > 0.005). No significant differences were found in the results of both treatments at 12 months, despite ALA exhibiting slightly better results at 3 months. No differences in pain and adverse events were assessed. Both ALA and MAL were effective when lower doses of red light were used in c-PDT. Long term efficacy was also documented. Further studies are necessary to determine the inferior point of red-light illumination without losing efficacy.

## 1. Introduction

Photodynamic therapy (PDT) is a non-surgical treatment for non-melanoma skin cancer, indicated in basal cell carcinoma and actinic keratossis (AK). PDT consist in the use of a photosensitizer to be selectively absorbed for the tumoural and premalignant cells, and afterwards destroy these cells with a convenience light source. The photosensitizers most widely used in cermatology are topical, which induce endogenous production of Protoporhirin IX. PPIX is activated by visible light and produces intracellular biological reactions in the tumoral cells via oxygen singlet production (ROS) and necrosis, leading to cellular death [1]. Throughout the decades, PDT has been used with different photosensitizers and light sources. Nowadays, conventional PDT in dermatology is known as the application of a topical photosensitizer, mostly aminolaevulinic acid (ALA) and metylaminolaevulinic acid (MAL), illuminated a red light LED lamp (680 nm, 37 J/cm^2^) [2].

Conventional photodynamic therapy (c-PDT) and daylight photodynamic therapy (DL-PDT) have been demonstrated to be effective and comparable treatments for multiple actinic keratosis (AK) [3]. Nevertheless, the difference in the doses of red light used between both modalities, which range from 37 J/cm^2^ to a lower total doses of red light in the visible light used for DL-PDT, suggests that maybe a lower dose of red light could be effective in c-PDT. The reason for exploring different forms of illumination in PDT are to relieve pain during the treatment without losing results. Red light-emitting diodes (LED) have shown superiority to other light sources, are the most used devices for performing PDT and are preferred by patients [4]. Optimizing the conventional lamp would be a possible approach to improve tolerance to PDT.

Undoubtedly, DL-PDT has emerged as a great alternative for illumination in PDT, even though there is still a lack of exploration of the influence of the light source and doses used in the global results of PDT. With this argument in mind, we performed a previous study comparing red light conventional illumination (Aktilite^®^, Galderma, Spain, 630 nm, 37 J/cm^2^) with half time illumination with MAL, obtaining similar results [5]. To point up, it seems that the optimal red light doses with the minimal patient discomfort need yet to be defined.

Both aminolaevulinic acid (ALA) and methylaminolaevulinic acid (MAL) have been demonstrated to be effective photosensitizers in c-PDT and DL-PDT obtaining similar results [3,4].

We performed a prospective, comparative, and blind study to assess the efficacy, tolerability, and safety of 17 J/cm^2^ of red light doses (h-PDT) for multiple actinic keratosis (AK) with aminolaevulinic acid (ALA) and methylaminolaevulinic acid (MAL) with long-term follow-up.

## 2. Materials and Methods

Patient candidates were selected for treatment if they had PDT with more than five symmetrically distributed AK of grade I or II on the scalp (Appendix A, Figure A1). The research was conducted between September of 2019 and December of 2021. The study was approved by the Ethics Committee of the hospital, and patients all signed informant consent. Patients were divided randomly into 7.8% ALA gel (Ameluz^®^, Biofrontera, Germany) or 16% MAL cream (Metvix^®^, Galderma, Spain) treatment. A nurse trained in PDT procedure, but not otherwise involved in the study, performed randomization.

Age, sex, and phototype of the patients were compiled at the basal visit. The number of total AK in the scalp were counted, mapped, drawn, and classified into grades I and II. Photographs of patients were taken, and AKASI was calculated.

Curettage of grade II AK was performed and dressed after the photosensitizer occlusion for three hours. The illumination (Aktilite^®^, 630 nm) was shortened into half, in time and doses, and 16 J/cm^2^ were applied for 4 minutes.

After PDT, patients evaluated the pain suffered in a visual analogue scale (VAS) from 0 to 10 and were instructed to completely avoid sun exposure in the treated areas for the next 48 hours. 

A questionnaire was given in the basal visit to be filled out at home 48 hours after PDT. Patients were instructed to subjectively evaluate the adverse effects from 0 to 3: erythema, edema, crusting, and blistering (0: not present; 1: light; 2: moderate; 3: severe). 

The next visits were scheduled 3 and 12 months later, in which AK were assessed. Patients were all evaluated by the same blinded dermatologist, who took no part in the treatment procedure. The primary endpoint was the complete clearance of each AK, and new lesions on the treated area were not evaluated at any time during the follow-ups.

AK complete clearance per patient was compared between groups using Student’s *t*-test and a 95% confidence interval (CI), assuming the independency between lesions within patients. The basal characteristics pre-treatment were compared between groups using the two-tailed Student’s t-test with a significance value of *p* < 0.05. For the statistical analysis of pain (VAS) and adverse effects, the ANOVA two-tailed test for independent data was used with a significance value of *p* < 0.05. 

## 3. Results

A total of 46 patients completed the study, 24 treated with MAL and 22 with ALA (Table 1). The median age of the patients treated was 77.63 and 80.14, respectively, and all of them were men with phototype II (fair skin). The distribution of AK was comparable in both groups (*p* > 0.005). The MAL group presented a medium basal AKASI of 6.51 (SD 1.17), a total of 27.13 AK per patient (15.88 grade I and 10.33 grade II), with a total of 651 lesions, divided into 391 grade I and 260 grade II. On the other hand, the ALA group had a medium AKASI of 6.81 (SD 1.51), with 30.95 AK per subject (19.23 grade I and 12.64 grade II), and a total of 681 lesions, divided into 413 grade I and 278 grade II.

The results of the comparison of the efficacy of both treatments are summarized in Table 2. After the treatment, both groups of patients improved, but with statistical differences; the ALA group achieved a better response (7.77 vs. 14.59, *p* = 0.016) at 3 months with a persistence of 25% of the overall AK (vs. 52% with MAL, *p* = 002). At 12 months, no statistical differences were found (*p* = 0.22) with a mean of lesion per patient of 13.80 with MAL (overall 46%) and 8.09 (overall 26%) with ALA.

Pain during PDT was similar in both groups, with a VAS of 5.21 for MAL and 5.31 for ALA (Table 3). With respect to the adverse events, edema and blisters reached zero punctuation, so they were not analysed. The presence of erythema and crusts were similar and low without statistical differences (Table 3). The VAS after treatment was near to none with no differences (*p* = 0.221).

## 4. Discussion

The reasons for exploring different light sources in PDT include obtaining extra benefits as intense pulsed light (IPL) or lasers in rejuvenation, shortening the time of illumination [6], decreasing pain [4], or simplifying the technique with LED [3]. A lot of new LED-based devices have proven efficacy [7], but the most practical approach is likely to optimize the most widely used traditional LED lamp.

Since PDT was first implemented, 37 J/cm^2^ were used in the c-PDT protocol (Aktilite^®^), despite the fact that they were painful and produced side effects such as erythema, crusts, and blisters [3,4]. Subsequently, the development of DL-PDT expanded the knowledge of illumination in a moderate way, and lower doses of red light were applied [5], demonstrating the same effectivity with more local damage (ROS production), and suggesting that the determinant factor for cytotoxicity is the total doses delivered, and not the irradiance or the PpIX accumulation [8]. In a comparative study modeling the local damage in PDT, DL-PDT achieved more local damage than C-PDT, suggesting that higher doses of red light are not related with higher lesion destruction [9] Moreover, when different protocols of illuminations are compared, the best option is likely the one with the best results at three months without pain [9], a remarkably practical approach.

Immunosuppression with high red light doses of PDT have been found after treatment, not only local but also systemic. It is worrisome that the capacity for fostering tumours in the treated area could be related [10]. Reaching a correct immune memory response is more beneficial [11]. These arguments are similar to the principles of photobiomodulation, which uses LED-light properties without a photosensitizer in modulating biological effects, and consequently, lower doses with conventional red light LED illumination could be as effective [5], non-suppressive, restorative, and less painful [12]. None of the patients treated in this study developed any malignancy during the follow-up period, nevertheless, it was a small sample. In the literature review, after the application of DL-PDT, no tumours had been described either [4,5,6,7,8,9,10,11,12,13].

In a previous study, half-time illumination (h-PDT) with c-PDT showed similar efficacy to c-PDT [5], and similar efficacy was achieved in h-PDT with ALA and MAL. Both groups of patients improved, achieving a maintenance response which was eventually not significant at 12 months. However, patients treated with ALA reached better significant results at 3 months (*p* = 0.016). The results in both groups, comparable at the baseline (*p* = 0.178, Table 1), were low in comparison with other studies in c-PDT [3,5,14] (54% with MAL and 74% with ALA overall reduction). Nevertheless, the severity of the patients treated with a mean basal AKASI higher than six (Table 1) and a mean of 27.13 AK per subject in the MAL group and 30.95 in the ALA group, with only one session applied, should be considered [15]. 

The side effects were mild, erythema being the most frequent with a medium punctuation of 1.3 out of 4 for MAL and 1.09 out of 4 for ALA (Table 3) without significant differences. Other side effects assessed had a very low score. There is no established protocol for evaluating side-effects after PDT [15], and the scale of our study was filled in by patients at home and was not validated with the consequent limitation.

With respect to pain, both photosensitizers exhibited similar punctuation in VAS, 5.21 with MAL and 5.31 with ALA on a scale from 0 to 10. The most frequent scale used to evaluate pain during PDT in the literature is the visual analogue scale (VAS) from 0 to 10, and c-PDT usually appeared with a mean of 4.4 to 5.7 in the VAS, and daylight was nearly painless [4,16,17]. Thus, h-PDT continued to be painful, the difference being that that pain lasts half the time. In a practical sense, if a patient had unbearable pain, the illumination could be shortened without losing efficacy if it had been at least a half.

## 5. Conclusions

In conclusion, both ALA and MAL are comparable and effective when 16 J/cm^2^ of red light in c-PDT is used. This protocol could be used to relieve pain during illumination. Further studies are necessary to truly assess if local immunosuppression is avoided and what the cost of effectivity is when illumination is shortened.

## Figures and Tables

**Table 1 biomedicines-10-03218-t001:** Patients and actinic keratosis (AK) characteristics at the baseline.

		MAL (*N* = 24)	ALA (*N* = 22)	*p* Value
Age		77.63 (±8.41)	80.14 (±4.93)	*p* = 0.229
Sex		24M/0F	22M/0F	NA
Phototype		2.13 (±0.34)	2.05 (±2.13)	*p* = 0.350
BASAL AKASI (AKASI_0_) *		6.51 (±1.17)	6.81 (±1.51)	*p* = 0.438
Total lesions per subject Mean ± SD		27.13 (±10.34)	30.95 (±8.42)	*p* = 0.178
AKP0* grade I		15.88 (±8.59)	19.23 (±6.82)	*p* = 0.152
AKP0* grade II		10.33 (±6.82)	12.64 (±5.82)	*p* = 0.316
Total lesions (*n*)	1332	651	681	*p* = 0.239
Grade I	804	391	413	
Grade II	532	260	278	

* AKASI_0_: Total AKASI of the sample at baseline expressed in mean and standard deviation; AKP0: Total count of AK per patient at baseline. Comparative groups *p* > 0.005.

**Table 2 biomedicines-10-03218-t002:** Summary of the comparative results per patient and total lesions.

	MAL (*N* = 24)	ALA (*N* = 22)	*p* Value *
BASAL			
Total basal AK/per subject (AKP0)	27.13 (±10.34)	30.95 (±8.42)	*p* = 0.178
Total AK = 1332	651	681	
3 MONTHS			
Total AK/per subject (AKP3)	14.59 (±11.32)	7.77 (±5.9)	*p* = 0.016
Total AK = 511	341/651 (52%)	171/681 (25%)	*p* = 0.02
12 MONTHS			
Total AK/patients at 12 months (AKP12)	13.80 (±9.15)	8.09 (±4.80)	*p* = 0.22
Total AK = 476	298/651 (46%)	178/681 (26%)	*p* = 0.244

* t: Student independent data.

**Table 3 biomedicines-10-03218-t003:** Comparison of the pain in a visual analogue scale (VAS, 0–10)) and local side effects (LSE).

	MAL (*N* = 24)	ALA (*N* = 22)	*p* Value *
VAS	5.21 (±2.3)	5.31 (±1.64)	*p* = 0.32
Erythema	1.3 (±0.48)	1.09 (±0.29)	*p* = 0.854
Edema	------	-------	NA
Crusts	0.25 (±0.61)	0.41 (±0.66)	*p* = 0.345
Blisters	------	-------	NA

* ANOVA two-tailed test independent data.

## Data Availability

Not applicable.

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
