# Peer review of "Methyl Aminolaevulinic Acid versus Aminolaevulinic Acid Photodynamic Therapy of Actinic Keratosis with Low Doses of Red-Light LED Illumination: Results of Long-Term Follow-Up"

_biomedicines, 2022, doi:10.3390/biomedicines10123218_

Round 1
Reviewer 1 Report
This report relates to the efficacy of a reduced light dose for treatment of AK with PDT. It was concluded that reducing the light dose by 50% did not affect efficacy. English usage is generally good with a few exceptions, e.g., ‘half-illumination’ and ‘inferior’ in the Abstract. A few more examples are noted elsewhere. ALA and MAL effects were examined over a 12-month interval.
Mention is made of use of daylight for PDT. The rationale for this is obscure. For regions without adequate irradiation facilities, this might be indicated. The connection between decreasing the controlled light dose and use of daylight is not clear. It seems unlikely that any group that needs to rely on daylight for PDT is going to have instruments that can measure the photon flux. While it may be feasible to reduce the light dose and still achieve a satisfactory outcome, it is not clear how this could affect treatment by ‘daylight’. This is a distraction. The pertinent element of this report is a study of the optimal light dose for maximum efficacy and minimal discomfort. The authors have chosen two light doses for this study: 16 and 37 J/sq cm. I suggest deleting such terms as ‘half-time’ and ‘half-doses’. Replace these with the actual light dose used.
The Tables use the comma rather than the period to designate the decimal point. I suggest using the period which will be more familiar to most readers, along with the symbol ± to designate the statistics. The first entry under MAL then becomes 77.63 ± 8.41.
Minor concerns: What is phototype II (line 83); what does ‘are variable’ mean (line 120); ‘fill-in’ (line 168); line 134 -change ‘have’ to ‘has’; line 123 change ‘were’ to ‘was’. What does ‘punctuation’ mean (line 156)? There are a few other examples where editing is needed.
Author Response
ANSWER TO REVIEWER 1
First, I would like to thank the reviewer for the time in improvement the manuscript. I believe the comment would enrich the manuscript. Then, I proceed to answer the revisions, all the changes are highlighted:
R1: This report relates to the efficacy of a reduced light dose for treatment of AK with PDT. It was concluded that reducing the light dose by 50% did not affect efficacy. English usage is generally good with a few exceptions, e.g., ‘half-illumination’ and ‘inferior’ in the Abstract. A few more examples are noted elsewhere. ALA and MAL effects were examined over a 12-month interval.
I proceed to correct. I eliminate “Half illumination” in the abstract and put “ lower dosed of red light”
R1: Mention is made of use of daylight for PDT. The rationale for this is obscure. For regions without adequate irradiation facilities, this might be indicated. The connection between decreasing the controlled light dose and use of daylight is not clear. It seems unlikely that any group that needs to rely on daylight for PDT is going to have instruments that can measure the photon flux. While it may be feasible to reduce the light dose and still achieve a satisfactory outcome, it is not clear how this could affect treatment by ‘daylight’. This is a distraction. The pertinent element of this report is a study of the optimal light dose for maximum efficacy and minimal discomfort. The authors have chosen two light doses for this study: 16 and 37 J/sq cm. I suggest deleting such terms as ‘half-time’ and ‘half-doses’. Replace these with the actual light dose used.
I completely agree with this comment, and believe it is very useful for improving the rational of this study. I change it in the introduction and the change are highlighted in the main document, anyway, I put bellow a copy:
“Conventional photodynamic therapy (c-PDT) and daylight photodynamic therapy (DL-PDT) have been demonstrated to be effective and comparable treatments for multiple actinic keratosis (AK)[1]. Nevertheless, the difference in the doses of red light used between both modalities, which range from 37 J/cm2 to a lower total dose of red light in the visible light used for DL-PDT, suggests that maybe lower doses of red light could be effective in c-PDT. The reason for exploring different forms of illumination in PDT are to relieve pain during the treatment without losing results. Red-light emitting diodes (LED) have shown superiority to other light sources, are the most used devices for performing PDT and are preferred by patients [2]. Optimize the conventional lamp would be a possible approach to improve tolerance to PDT.
Undoubtedly, DL-PDT has emerged as a great alternative for illumination in PDT, even though there is still a lack of exploration of the influence of the light source and doses used in the global results of PDT. With this argument in mind, we performed a previous study comparing red light conventional illumination (AktiliteÒ, 630 nm, 37 J/cm2) with half time illumination with MAL, obtaining similar results [3]. To point up, it seems that the optimal red light doses with the minimal patient discomfort need yet to be defined.
Both aminolaevulinic acid (ALA) and methylaminolaevulinic acid (MAL) have been demonstrated been effective photosensitizers in c-PDT and DL-PDT obtaining similar results [1,2]
We performed a prospective, comparative, and blind study to assess the efficacy, tolerability, and safety of 17J/cm2of red light doses (h-PDT) for multiple actinic keratosis (AK) with aminolaevulinic acid (ALA) and methylaminolaevulinic acid (MAL) with long-term follow-up.
R1: The Tables use the comma rather than the period to designate the decimal point. I suggest using the period which will be more familiar to most readers, along with the symbol ± to designate the statistics. The first entry under MAL then becomes 77.63 ± 8.41.
Thank you so much for this correction, my local insights appeared in this way of presenting the tables. I would change it.
Minor concerns: What is phototype II (line 83); what does ‘are variable’ mean (line 120); ‘fill-in’ (line 168); line 134 -change ‘have’ to ‘has’; line 123 change ‘were’ to ‘was’. What does ‘punctuation’ mean (line 156)? There are a few other examples where editing is needed.
Phototype II: I add in brackets (fair skin)
Are variable: I eliminated that, the phrase change to: “The reasons for exploring different light sources in PDT include obtaining extra benefits as intense pulsed light (IPL) or lasers in rejuvenation, shortening the time…”
Fill-in: I suppose the suggested change is “Fill in”, not “fill out”
I change all the words suggested, punctuation for score. I proceed to deeply reviewing the text.
Thank you so much again for your work and your time.

Reviewer 2 Report
Guarino and co-worker present in their submission to Biomedicines "Methyl aminolaevulinic acid versus aminolaevulinic acid photodynamic therapy of actinic keratosis with low doses of red-light LED illumination: results of long-term follow-up." The following issues must all be carefully addressed and the revised manuscript must be checked again.
The following reference should be added: J Photochem Photobiol B 2017 174:70.
"Nevertheless, the difference in the doses of red light used between both modalities, which range from 37 J/cm2 to a minimum of 8 J/cm2 in daylight, suggests that intermediate doses could be effective." Reference(s) for this sentence must be given, since the 8 J/cm2 is not mentioned in ref. 1.
"cm2": the "2" must be superscript (several times).
"DL-PDT depends on the weather" This is a very simplified view. It also depends upon the daytime, altitude, season and geographical location.
"Patient were divided randomly into ALA (Ameluz, Biofrontera, Germany) or MAL (Metvix, Galderma, España) treatment." It is never mentioned, what was the used dose of ALA or MAL.
After "SD" there should always be a space.
"supplementary material S2" This material is missing.
Ref. 5 is formatted differently from the rest.
Author Response
ANSWER TO REWIEVER 2
Firstly, I would like to thank the reviewer for his work and the time expended in improving the article. Then, I proceed to answer one by one the issues.
R2: Guarino and co-worker present in their submission to Biomedicines "Methyl aminolaevulinic acid versus aminolaevulinic acid photodynamic therapy of actinic keratosis with low doses of red-light LED illumination: results of long-term follow-up." The following issues must all be carefully addressed and the revised manuscript must be checked again.
R2:The following reference should be added: J Photochem Photobiol B 2017 174:70.
Then, I proceed to answer one by one the issues. I believe it is very interesting, I added it, with a comment in the text, it is reference 7.
R2:"Nevertheless, the difference in the doses of red light used between both modalities, which range from 37 J/cm2 to a minimum of 8 J/cm2 in daylight, suggests that intermediate doses could be effective." Reference(s) for this sentence must be given, since the 8 J/cm2 is not mentioned in ref. 1.
I have eliminated that part of the discussion as other reviewer suggested. The changes are highlighted.
"cm2": the "2" must be superscript (several times).
I correct it
"DL-PDT depends on the weather" This is a very simplified view. It also depends upon the daytime, altitude, season and geographical location.
I have eliminated that part of the discussion as other reviewer suggested. The changes are highlighted.
"Patient were divided randomly into ALA (AmeluzÒ, Biofrontera, Germany) or MAL (MetvixÒ, Galderma, España) treatment." It is never mentioned, what was the used dose of ALA or MAL.
I suppose that you are referring to the quantity in the cream? I add that quantity.
After "SD" there should always be a space.
The other reviewer changed it into ±
"supplementary material S2" This material is missing.
I eliminated that sentence; it was a mistake.
Ref. 5 is formatted differently from the rest.
I correct it.
Thank you so much again for your time and your work.
The author